# An Increase of GNSS Data Time Rate and Analysis of the Carrier Phase Spectrum

**Vladislav Demyanov** [1,2,*] **, Ekaterina Danilchuk** [3] **, Maria Sergeeva** [4,5] **and Yury Yasyukevich** [1]

1   Institute of Solar-Terrestrial Physics, Siberian Branch of Russian Academy of Sciences, Irkutsk 664033, Russia
2   Department of Automatic and Telecommunication, Irkutsk State Transport University, Irkutsk 664074, Russia
3   Department of Radio-Wave Physics and Radio-Engineering, Irkutsk State University, Irkutsk 664003, Russia
4   SCiESMEX, LANCE, Instituto de Geofisica, Unidad Michoacan, Universidad Nacional Autonoma de Mexico, Antigua Carretera a Patzcuaro 8701, Morelia 58089, Michoacan, Mexico
5   CONACYT, Instituto de Geofisica, Unidad Michoacan, Universidad Nacional Autonoma de Mexico, Antigua Carretera a Patzcuaro 8701, Morelia 58089, Michoacan, Mexico
*   Correspondence: vv.emyanov@gmail.com; Tel.: +7-950-051-3095

**Abstract:** Natural hazards and geomagnetic disturbances can generate a combination of atmospheric and ionospheric waves of different scales. The carrier phase of signals of global navigation satellite system (GNSS) can provide the highest efficiency to detect and study the weak ionospheric disturbances in contrast to total electron content (TEC) and TEC-based indices. We consider the border between the informative part of the carrier phase spectrum and the uninformative noises—the deviation frequency—as the promising means to improve the GNSS-based disturbance detection algorithms. The behavior of the deviation frequency of the carrier phase spectra was studied under quiet and disturbed geomagnetic conditions. The results showed that the deviation frequency value increases under magnetic storms. This effect was revealed for all GNSS constellations and signals regardless the GNSS type, receiver type/make and data rate (50 or 100 Hz). For the 100 Hz data, the most probable values of the deviation frequency grouped within ~28–40 Hz under quiet condition and shifted to ~37–48 Hz during the weak geomagnetic storms. Additionally, the lower values of deviation frequency of ~18–25 Hz almost disappear from the distribution of the deviation frequencies as it becomes narrower during geomagnetic storms. Considering that the small-scale irregularities shift the deviation frequencies, we can use this indicator as a "red alert" for weakest small-scale irregularities when the deviation frequency reaches ~35–50 Hz.

**Keywords:** ionosphere; scintillations; carrier phase; GNSS; GPS; GLONASS; Galileo; SBAS; GNSS signals; deviation frequency

## 1. Introduction

GNSS measurements provide big data sets of ionospheric observations. These observations may be used to study the ionospheric response to natural hazards [1] such as earthquakes [2], tsunamis [3], cyclones [4], volcanoes [5], and meteoroids [6,7] as well as rocket launches [8] and explosions [9]. These efforts mainly aim at the practical realization of the modern and future natural hazard detection and warning systems, for example GUARDIAN (A Near Real-Time Ionosphere Monitoring System for Natural Hazards Early Warnings) [10]. However, the systems like GUARDIAN, SIMuRG (system for ionosphere monitoring and research from GNSS) [11] and others use slant TEC time series as a base parameter to detect and explore ionospheric disturbances caused by natural and anthropogenic impacts.

The sensitivity of TEC as an ionospheric parameter and its derivatives (such as rate of TEC increasing (ROTI), derivative of ROTI (DROTI) et al.) is one of the main challenges to build up effective near real-time and post-processing ionosphere monitoring systems.

As TEC-based data are a product of inter-frequency ionosphere-free combinations, they can consist of higher noises and additional anomalies. First, there is significant difference between the L2P(Y) and L2C-derived TEC. The independently tracked carrier phase is more precise compared to the L1-aided observables, which eventually results in the more precise TEC values [12]. Second, the correlation between L1 and L2 carrier phase noises affects the total noise figure in TEC data [13].

An exact analysis of the GNSS carrier phase spectrum at different GNSS frequencies provides more accurate information about the small-scale structure of the ionosphere [14] and, therefore, may be used for effective studying of weak ionospheric disturbances [7]. We consider this paper as a continuous improvement of GNSS sounding methods to study the ionosphere based on GNSS high time rate data processing. It provides new opportunities to achieve better results in the ionosphere studying and higher sensitivity of the Ionospheric Monitoring Systems.

The manuscript improves our previous results [13,15]. The present paper is devoted to a problem of how the data time rate, GNSS signal type and geomagnetic activity affect the border between the informative part of the carrier phase spectra and its uninformative noisy part. In addition, we try to identify the threshold for detection of weak small-scale ionospheric disturbances depending on the above mentioned factors. To solve these tasks, we used 50 Hz and 100 Hz carrier phase data of GNSS L1, L2 and L5 frequencies from Javad and Septentrio receivers under both geomagnetically quiet and disturbed conditions.

## 2. Deviation frequency and GNSS Sounding Methods Sensitivity

Natural processes of different kinds can form atmospheric waves and their corresponding ionospheric disturbances of different spatiotemporal scale, lifetime and travelling velocities and directions. Large-scale disturbances are characterized by velocities of about 0.8–3.0 km/s and live for about an hour [16–18]. Small-scale disturbances usually have lower velocities and live longer, up to ~3 h [19]. It is rather common that some of the natural hazards trigger the low amplitude waves that are difficult to detect [6]. If this is the case, the sensitivity of the detection tool (the chosen parameter/index) takes on a great importance.

TEC-based parameters/indices are not always capable of the required sensitivity. The GNSS carrier phase is the most precise parameter among GNSS observables. Stochastic techniques report the normally distributed carrier phase noise of 0.002 m, while the code range noise is of (0.5–0.8) m [20].

The noise of the phase measurements is the main limitation for sensitivity of the GNSS sounding methods. McCaffrey and Jayachandran [21] suggested the "deviation frequency" term to denote the boundary between the variable part of the amplitude and phase variations spectrum and the uninformative noise in this spectrum. The deviation frequency (*fd*) value is found as a point in the phase variations spectra at which the spectral slope shallows to the near zero inclination.

Figure 1 illustrates *fd* that divides the informative part of the phase variations spectra and phase uninformative noise. One can see that the higher the *fd*, the larger the part of the spectrum is informative, providing more chances to detect the weak disturbances in the ionosphere. The sensitivity threshold of the GNSS sounding in this case corresponds to the beginning of the noise part of the spectrum where no ionospheric disturbance can be revealed. This threshold corresponds to the particular *fd* value and is marked by a blue horizontal line in Figure 1.

It was shown previously that the carrier phase noise depends on the GNSS receiver type/make [13], signal component (L1, L2C, L5, etc.) [22,23] and GNSS data temporal resolution [24]. These factors define the sensitivity threshold of the GNSS sounding methods used for the ionospheric studies and have to be taken into or consideration in this study.

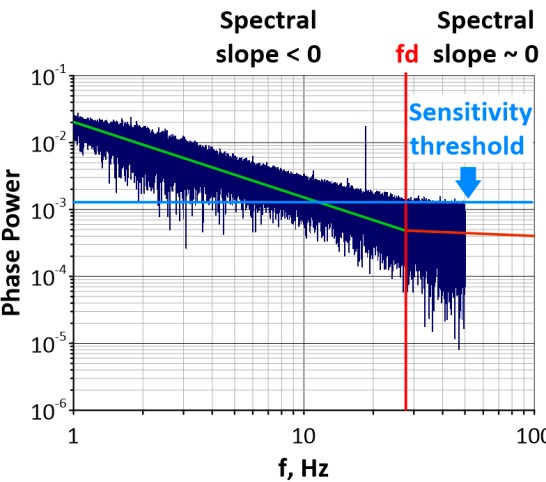

**Figure 1.** Deviation frequency and the sensitivity threshold of the GNSS sounding methods.

### 3. Experiment Description

We used data from two types of receivers: Javad Delta-G3T [25] and Septentrio PolaRx5 [26] belonging to SibNet network [27]. Both receivers are connected to the same RingAnt-G3T antenna through the antenna switcher. The equipment is installed in the Institute of Solar-Terrestrial Physics, Irkutsk, Russia (52°N, 104°E). The receivers can track signals of Global Positioning System (GPS), Global Navigation Satellite System (GLONASS), Galileo (European Satellite navigation System) and Space Based Augmentation Systems (SBAS) at L1, L2 and L5 frequencies. The antenna provides a signal pass in all these frequency bands and has the averaged slope about 9 dB/MHz [28]. Javad measurements were performed with the 50 Hz sampling rate and Septentrio with the 50 and 100 Hz sampling rates.

Table 1 summarizes the characteristics of GNSS signal components involved into analysis. The detailed description of the signal components and their characteristics can be found in the interface control documents of GPS and SBAS [29,30], GLONASS [31,32], and Galileo [33]. The Receiver Independent Exchange (RINEX) data format and the corresponding definition of the signal components are described in [34]. The details on the applied procedure of the carrier phase data processing can be found in [14].

**Table 1.** The GNSS signals and range codes characteristics.

| System | Signal | PRN Code Chip Rate, Mchips/s | PRN Code LengthChips | Total Received Minimum Power, dBW | Receiver Reference Bandwidth, MHz | Modulation Type |
|---|---|---|---|---|---|---|
| **GPS** | L1C | 1.023 | 1023 | −158.50 | 20.46 | BPSK |
| | L2C | 0.511 | 10,230 (CM code) 767,250 (CL code) | −164.5 (SV II) −160.0 (SV IIF) −158.5 (SV III) | 20.46 (30.69 for SV III Blocks) | BPSK |
| **GLONASS** | L5Q | 10.23 | 10,230 | −157.90 | 24.00 | BPSK |
| | L1C (CDMA) | 0.511 | 4092 | −158.50 | 17.10 | BPSK |
| | L1C (FDMA) | 0.511 | 511 | −161.00 | 8.00 | BPSK |
| | L2C (CDMA) | 0.511 | 10,230 | −158.50 | 19.00 | BPSK |
| | L2C (FDMA) | 0.511 | 511 | −161.00 | 7.00 | BPSK |
| **Galileo** | L1C | 1.023 | 4092 | −157.25 | 24.55 | CBOC |
| | L6C | 5.115 | N/A | −155.25 | 40.90 | CBOC |
| | L7Q | 10.230 | 10,230 | −155.25 | 20.46 | AltBOC |
| | L8Q | 10.230 | 10,230 | −155.25 | 20.46 | AltBOC |
| | L5Q | 10.230 | 10,230 | −155.25 | 20.46 | AltBOC |
| **SBAS** | L1C | 1.023 | 1023 | −161.00 | 24.00 | BPSK |

PRN—pseudo random noise (code); CM—civil moderate (code); CL—civil long (code); SV—satellite vehicle; BPSK—bi-phase shift keying (modulation); CBOC—composite binary offset (modulation); AltBOC—alternative BOC (modulation); CDMA—code division multiple access; FDMA—frequency division multiple access.

The obtained results were summarized in a form of histograms of the deviation frequency distribution (Figures 3–6). All the spectra under consideration generally have power law form which does not contradict the known theory [35]. Different ranges on each spectrum may have different slopes and it complicates the task of the *fd* identification. To build up the histograms of the *fd* distributions we analyzed ~ 4000 phase variation spectra (please see the Data Availability Statement for the paper below). This total amount of the spectra includes three sets of the data: 50 Hz and 100 Hz data of Septentrio and 50 Hz data of Javad. Each data set contains 2 days of measurements. In turn, each day consists of 24 one-hour data series and each one-hour set contains 35–40 GNSS satellites in view. To obtain our results, we considered each particular spectrum and define an individual deviation frequency within the expected frequency band of 10–50 Hz [21]. We did not consider spectra of complex form and spectra consisting of anomalies and relating to low elevation satellites.

Magnetic storms can significantly influence the form and features of the phase variations spectrum. According to [36], during the storms, the slope of the spectra and the intensity of TEC variations increase within the whole spectra (including small-scale weak disturbances). Hence, a composition of the high frequency part of the spectra and, correspondingly, the deviation frequency are both dependent on geomagnetic conditions.

Three weak geomagnetic storms that occurred on 16 April 2021 ($Dst_{min}$ = −48 nT), 10 April 2022 ($Dst_{min}$ = −44 nT), and 4 September 2022 ($Dst_{min}$ = −77 nT) were chosen for the analysis. 13 April 2021, 8 April 2022, and 2 September 2022 were quiet reference days. Figure 2 illustrates geomagnetic activity indices under quiet conditions (upper panels) and during the storms (lower panels).

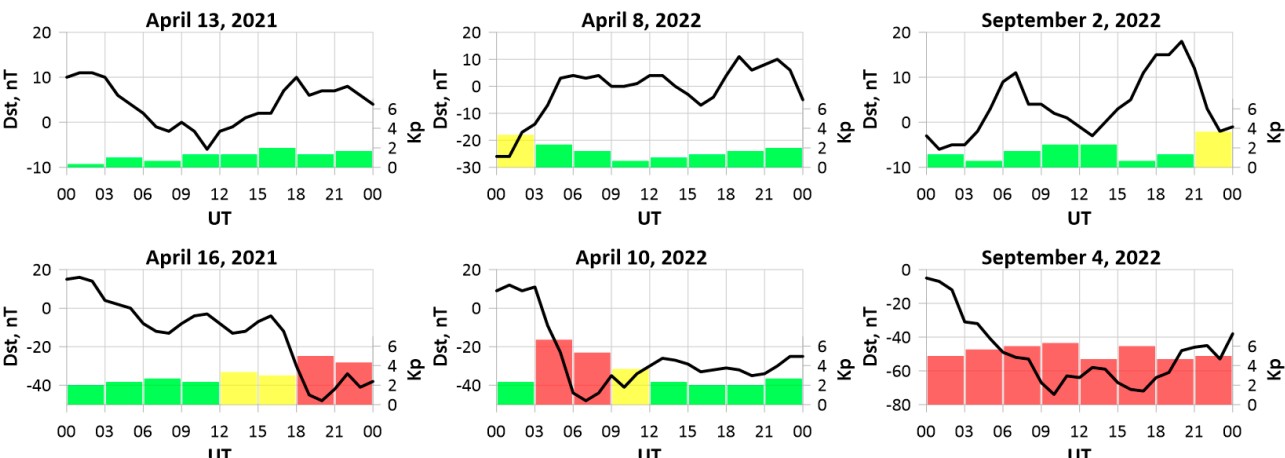

**Figure 2.** Kp (colored bars) and Dst (black curves) during the chosen storms (upper panels) and the quiet reference days (lower panels).

We studied two frequency bands in the carrier phase spectra in which the deviation frequency was expected to vary. To analyze the 50 Hz data, the frequency band from 10 to 25 Hz in the spectrum was considered [14]. In the case of the 100 Hz data, we considered the frequencies from 10 to 50 Hz [21].

Correlation properties of the particular GNSS signal component depend on the signal modulation type translating an individual satellite PRN code. The authors of [37,38] demonstrated that the new modulation types of navigation signals, such as AltBOC, CBOC, and Time Multiplexed BOC (TMBOC) bring potential improvements, for instance higher signal power and better multipath mitigation capabilities.

To analyze the *fd* behavior in the phase variations spectra, several factors defining the phase noise pattern should be considered: multipath, receiver thermal noise and correlation losses. All of them depend on the signal modulation type and characteristics of PRN code.

First, we discuss the multipath problem. In the simplest case, when there is one direct and one reflected signal, the multipath-caused phase measurement error may be calculated as follows [39]:

$$\delta\varphi = arctan\left(\frac{\alpha_1 \cdot R(\tau - \tau_1) \cdot sin\varphi_1}{R(\tau) + \alpha_1 \cdot R(\tau - \tau_1) \cdot cos\varphi_1}\right),$$

(1)

where $R(\tau)$ is an auto-correlation function of the direct signal; $R(\tau - \tau_1)$ is a cross-correlation function between the direct and the reflected signals; $\tau$ is the estimate for the receiver code pseudo-delay; $\tau_1$ is the reflected signal delay relative to the direct signal delay; $\alpha_1$ is the reflection coefficient; and $\varphi_1$ is the reflected signal phase.

The autocorrelation function is completely defined with the signal modulation type and PRN code characteristics (code chip length, chipping rate and the length of a PRN code sequence). Thus, Equation (1) demonstrates that the value of a multipath-caused error and its behavior differs from one GNSS signal component to another depending on the modulation type and PRN code structure (Table 1).

Next, it is known that thermal noise error in a signal phase tracking depends on the carrier-to-noise ratio (*CNR*) at the phase lock loop (*PLL*) input as follows [40]:

$$\sigma_\varphi = \sqrt{\frac{\Delta F_{PLL}}{CNR} \cdot \left(1 + \frac{1}{2T \cdot CNR}\right) + \left(m \cdot \frac{\sigma_F(\tau) \cdot f}{\Delta F_{PLL}}\right)^2}, \text{ rad}$$

(2)

where $\Delta F_{PLL}$ is the *PLL* noise bandwidth, Hz; *CNR* is the carrier to noise ratio, dBW; *T* is the integrate/damp period, ms; $\sigma_F(\tau)$ is the short-time instability of the receiver reference oscillator; $f$ is the signal carrier frequency, Hz; and $m$ is the coefficient depending on the lock lop order.

Note that the *CNR* depends on two parameters of the *PRN* code, the *PRN* code chip rate and the form of the *PRN* code autocorrelation function. The higher code chip rate yields the larger signal processing gain in the carrier-to-noise ratio as follows [40]:

$$\Delta CNR = 10 \cdot lg\left(\frac{F_{PRN}}{2 \cdot \Delta F_{DLL}}\right), \text{ dBW}$$

(3)

where $F_{PRN}$ is the code chip rate of a *PRN* code sequence (Mchips/sec) and $\Delta F_{DLL}$ is the code delay lock loop (*DLL*) noise bandwidth, Hz.

The form of the PRN code autocorrelation function defines the *CNR* at the *PLL* input and depends on errors of the code delay and carrier frequency computation as follows [41]:

$$CNR_E = CNR \cdot R^2(\varepsilon_\tau) \cdot sinc^2\left(0.5 \cdot \varepsilon_f \cdot T\right), \text{ dBW}$$

(4)

where $R(..)$ is the autocorrelation function for a *PRN* code; $\varepsilon_f$ is the signal frequency evaluation error, Hz; $\varepsilon_\tau$ is the code delay evaluation error, code chips; and *CNR* is the carrier to noise ratio at the *PLL* input with allowance for the receiver thermal noise only (2).

In addition, the length of a *PRN* code sequence can impact the *CNR*, too. In general, *CNR* value is considered as a relation between the averaged signal power $S^2(f)$ and the noise power $\sigma_N^2$ within the front-end bandwidth of the receiver as follows:

$$CNR = \frac{\langle S^2(f)\rangle}{\sigma_N^2}.$$

(5)

It is known that the line spacing of the *PRN* sequence line spectrum ($\Delta F$, Hz) decreases with a *PRN* code sequence length (*N*, chips) as follows [40]:

$$\Delta F = \frac{1}{N \cdot \tau_C}, \text{ Hz}$$

(6)

where $\tau_C$ is a code chip length, sec.

From (5) and (6), one may conclude that the averaged signal power of the signal modulated by *PRN* code of $N_1$ chips is higher than the signal of the $N_2$ code length if $N_1 > N_2$ for the same signals accumulate/damp period.

Equations (1)–(6) prove that better correlation properties of the *PRN* code, higher PRN code chip rate and longer *PRN* code sequence help to reduce multipath noise and achieve a higher *CNR* and a higher accuracy in the signal phase estimate. Therefore, different GNSS signal components are expected to have different signal phase spectra and different variations of the deviation frequencies. This emphasizes the importance of the analysis of behavior of the deviation frequency for signals of different navigation systems, different frequencies and signal components separately.

## 4. Experimental Results

### 4.1. Dependence of the Deviation Frequency on the Sampling Rate

Figure 3 shows histograms of *fd* distribution depending on the data sample rate. Please note that only the results for the civil L1 signal (L1C) are shown in this figure for the economy of space. Exact analysis depending on the GNSS signals is provided in the next section.

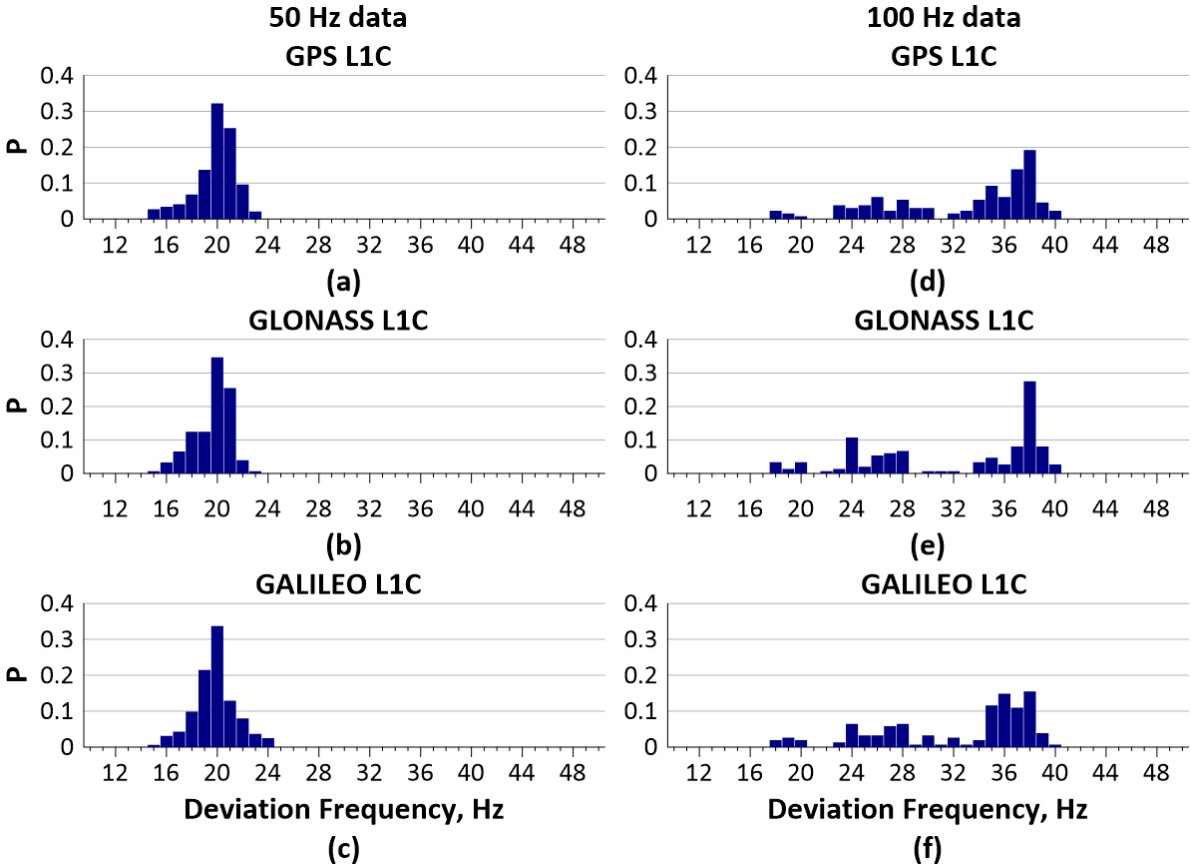

**Figure 3.** The *fd* distributions on 2 September 2022 for L1C signals of 50 Hz (left panels) and 100 Hz data (right panels), GPS (**a**,**d**), GLONASS (**b**,**e**) and Galileo (**c**,**f**). *P* is the normalized probability of the *fd* distribution.

To exclude the effects of magnetic storm, the presented distributions were obtained for quiet geomagnetic conditions and for all GNSS satellites in view (excluding satellites whose elevation angle was below 30° to mitigate the multipath influence).

A common feature is seen for both 50 and 100 Hz sample rates. The deviation frequencies are mostly grouped closer to the Nyquist frequency defined by the digitization frequency ($f_{dg}$) according to the Kotelnikov's theorem as $f_{dg} = 2F_{Nq}$. The deviation

frequencies mostly group within ~35–42 Hz for 100 Hz data and within ~18–24 Hz for the 50 Hz.

According to Figure 3, the increase in the data sampling rate significantly expands the band of deviation frequencies. Regardless the GNSS constellation, three bands of *fd* are seen in Figure 3: ~17–21, 23–30 and 32–42 Hz. This demonstrates that a higher sampling rate provides better opportunities for the weaker disturbance detection. In contrast, the 50 Hz data shows a narrow band of the deviation frequencies, ~17–24 Hz. This means that 50 Hz data does not provide many opportunities to detect weak, small-scale ionospheric turbulences which lay beyond Nyquist frequency in the noise tail of the phase variations spectra. This will be considered in more details in the Discussion section.

### 4.2. Variations of Deviation Frequency Depending on Geomagnetic Conditions

Figure 4 shows *fd* distributions of the deviation frequencies obtained from measurements based on data from two different receivers under quiet conditions (first and third columns) and during geomagnetic storms (second and forth columns). Similarly to Figure 3, only results obtained from L1C signal processing are shown.

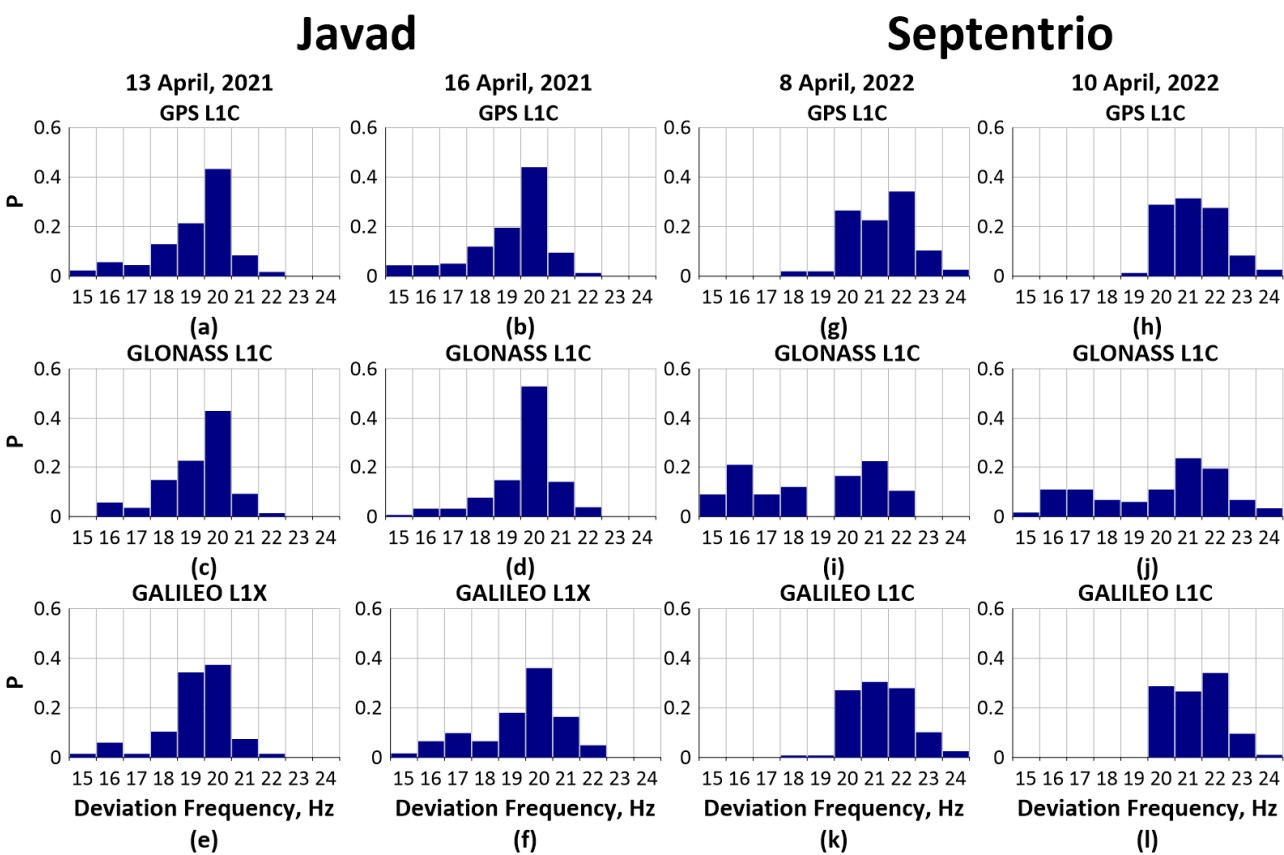

**Figure 4.** Distribution of the deviation frequencies obtained from 50 Hz data by Javad (**a**–**f**) and Septentrio (**g**–**l**) under different geomagnetic conditions.

The highest *fd* value by Javad data is lower (22 Hz) than the value by Septentrio (24 Hz). During the storms, *fd* values by data of both receivers were higher than during quiet periods. This feature seems much more pronounced for Septentrio but it is not very significant for Javad data.

Figure 5 shows results of the same distributions but based on data for all signal components of all GNSS constellations by Septentrio receiver.

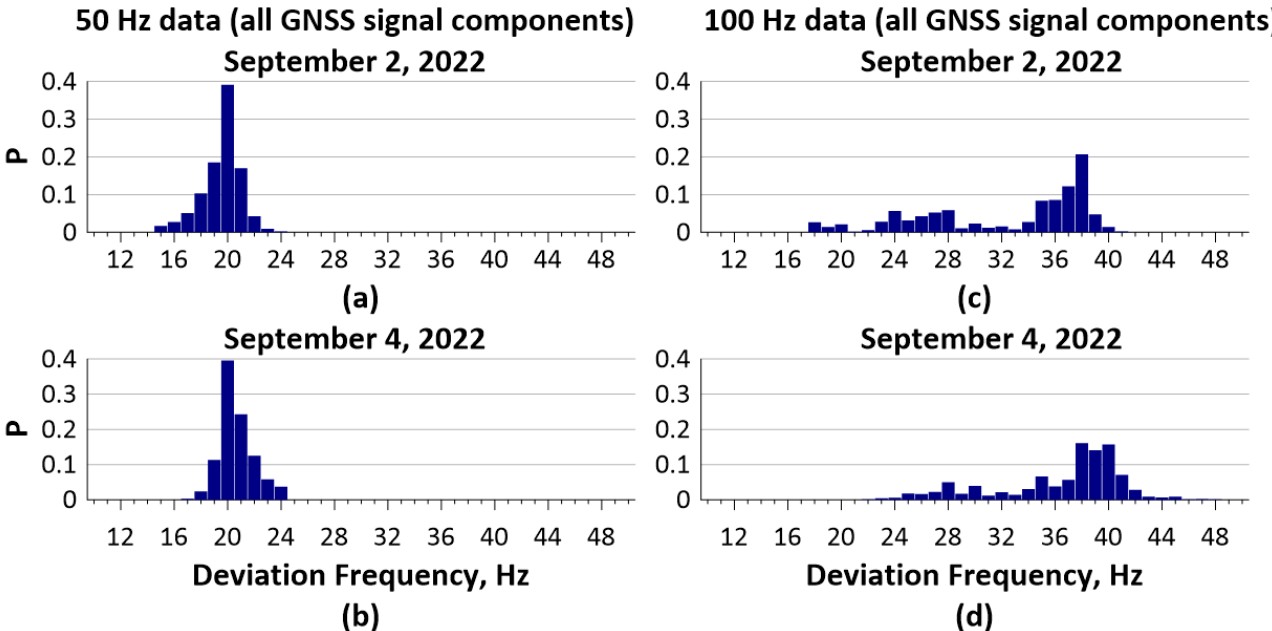

**Figure 5.** The deviation frequency distribution obtained from 50 Hz (**a**,**b**) and 100 Hz (**c**,**d**) data of all GNSS signal components by Septentrio receiver under quiet conditions (**a**,**c**) and under magnetic storm (**b**,**d**).

One can see that the higher *fd* values are present under disturbed conditions for both 50 Hz and 100 Hz data sets. The higher deviation frequencies reach ~35–40 Hz under quiet conditions (Figure 5c) and ~38–46 Hz (Figure 5d) during the storm. Another feature is that *fd* values within ~18–22 Hz are almost absent during the storm. Due to this, the distribution bands narrow and shift to the higher frequencies in contrast to distributions for the quiet conditions.

The 50 Hz data showed much weaker response to the magnetic storm (Figure 5a,b). Nevertheless, the most probable deviation frequency of 20 Hz in both histograms is the same under quiet and disturbed conditions. At the same time, the form of histogram changes implying some shift to higher frequencies (Figure 5b).

Both cases obviously allow us to draw the conclusion that deviation frequencies obviously react on magnetic storm. Magnetic storm causes drift of the *fd* values to higher frequencies regardless of the data sampling rate.

*4.3. Maximal Observed Deviation Frequency Depending on the GNSS Signal Component*

Table 2 summarizes the results obtained in this study for 50 Hz data from different GNSS constellations and their signal components under quiet conditions and during geomagnetic storms. The data were obtained from the Septentrio receiver on April 2022 and September 2022. The same data set obtained from the Javad receiver on April 2021 is presented on Table 3.

Based on Tables 2 and 3 the following conclusions are made.

1.  Without regard to the particular GNSS system or its signal component, *fd* reacts to a geomagnetic storm. It manifests as the shift of *fd* values to higher frequencies in the majority of cases.
2.  Different signal components show different intensities of response. This difference is significantly more pronounced for the Septentrio data set. This is probably due to the lower phase measurement noise compared to the phase measurement noise in the Javad receiver [13].
3.  The *fd* value for GPS and Galileo signals varied slightly more than for GLONASS and SBAS.

**Table 2.** The deviation frequencies obtained from Septentrio receiver at 50 Hz sampling rate.

| System | Signal | Observed Deviation Frequencies, Hz | | | |
| --- | --- | --- | --- | --- | --- |
| | | Quiet Conditions | Magnetic Storm | Quiet Conditions | Magnetic Storm |
| | | September 2 | September 4 | April 8 | April 10 |
| GPS | L1C | 15–23 | 17–24 | 18–24 | 19–24 |
| | L2C | 15–22 | 18–24 | 19–24 | 18–24 |
| | L5Q | 15–22 | 18–24 | 16–24 | 19–24 |
| GLONASS | L1C | 15–23 | 18–24 | 15–22 | 15–24 |
| | L2C | 15–23 | 18–24 | 15–22 | 16–24 |
| Galileo | L1C | 15–24 | 17–24 | 18–24 | 20–24 |
| | L6C | 15–23 | 17–24 | 18–24 | 18–24 |
| | L7Q | 15–22 | 17–24 | 18–24 | 18–24 |
| | L5Q | 15–22 | 18–24 | 19–24 | 18–24 |
| SBAS | L1C | 15–23 | 15–24 | 20–24 | 20–24 |

**Table 3.** The deviation frequencies obtained from Javad receiver at 50 Hz sampling rate.

| System | Signal | Observed Deviation Frequencies, Hz | |
| --- | --- | --- | --- |
| | | Quiet Conditions | Magnetic Storm |
| GPS | L1C | 15–22 | 16–22 |
| | L1W | 15–22 | 16–22 |
| | L2W | 15–22 | 16–22 |
| | L2X | 15–22 | 15–22 |
| | L5X | 15–22 | 15–22 |
| GLONASS | L1C | 16–22 | 15–22 |
| | L1P | 16–22 | 15–22 |
| | L2C | 15–22 | 15–22 |
| | L2P | 15–22 | 16–22 |
| Galileo | L1X | 15–22 | 15–22 |
| | L5X | 15–21 | 15–22 |
| SBAS | L1X | 13–20 | 15–20 |

Considering the issues (1–3), it may be concluded that the receiver hardware noise and data sampling rate can limit the sensitivity of GNSS sounding methods for weak ionospheric disturbance detection. More features of *fd* dependence on the signal component can be found with the higher temporal resolution (100 Hz). Figure 6 illustrates the examples of the significant difference in maximal observed *fd* values for two GPS signal components under the same conditions.

Both L2L and L5Q carrier components are bi-phase shift key (BPSK)-modulated (Table 1). The Q5 code has the 10.23 MHz chip rate and the 1-ms duration (one PRN sequence contains 10,230 bits). The CL-code has the 511.5 kHz chip rate but its duration is much longer and is equal to 1.5 s (one PRN sequence contains 767,250 bits). The L5Q signal has a higher code chip rate compared to the L2L signal. However, the L2L signal component demonstrates much higher maximal observed *fd* values (Figure 6b). This probably proves the better sensitivity of the L2L signal component to the smaller and weaker ionospheric disturbances triggered by the geomagnetic storm. We think it is due to the much longer CL code, which results in the CNR improvement (Equations (5) and (6)), and lower noise errors of tracking, which means higher sensitivity of the GNSS-based methods for the ionospheric studies.

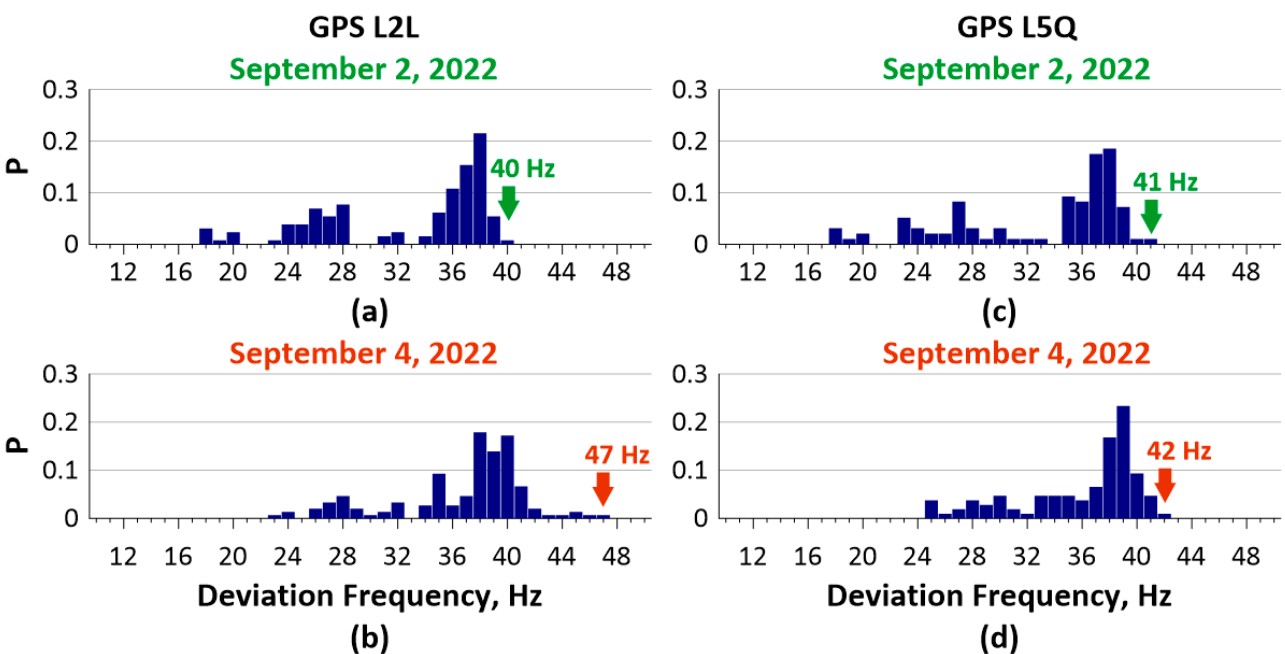

**Figure 6.** The deviation frequency fd distribution for 100 Hz data of L2L and L5Q GPS signals under quiet conditions on 2 September 2022 (**a**,**c**) and during the magnetic storm on 4 September 2022 (**b**,**d**).

Let us summarize all the received in this study results. Table 4 contains the detailed information on *fd* variations obtained from different GNSS signals by the Septentrio receiver with the 100 Hz data sampling rate.

**Table 4.** The observed deviation frequencies obtained from GNSS signals at 100 Hz by Septentrio receiver.

| System | Signal | Observed Deviation Frequencies, Hz | |
|---|---|---|---|
| | | Quiet Conditions (2 September 2022) | Magnetic Storm (4 September 2022) |
| GPS | L1C | 18–40 | 23–44 |
| | L2C | 18–40 | 23–47 |
| | L5Q | 18–41 | 25–42 |
| GLONASS | L1C | 18–40 | 22–43 |
| | L2C | 18–40 | 22–43 |
| GALILEO | L1C | 19–40 | 25–48 |
| | L6C | 18–40 | 23–46 |
| | L7Q | 19–41 | 24–45 |
| | L8Q | 18–41 | 25–45 |
| | L5Q | 19–40 | 23–48 |
| SBAS | L1C | 18–40 | 20–41 |

We already mentioned the apparent *fd* reaction to the magnetic storms regardless of the GNSS constellation and signal component. Table 4 confirms this conclusion and demonstrates that the *fd* values increase for all GNSS signal components under magnetic storm conditions. To summarize, our results imply the higher sensitivity of the particular signal components to detect the weak ionospheric disturbances. The Galileo signal components seem more promising in this regard. The probable explanation for this is the more sophisticated modulation type which provides much lower a multipath phase tracking errors and better carrier-to noise ratio.

## 5. Discussion

Correct definition of the *fd* value is a difficult task and we have to define it at each spectrum reliably by means of visual analyzing. However, our evaluations of the deviation frequency did not contradict known results [21], which demonstrated very wide peaks of the deviation frequencies, spanning a portion of the 20–30 Hz and 30–40 Hz windows.

Due to an impact of high frequency components of ionospheric phase variations, the maximal frequency of the spectrum ($F_{max}$) is unknown and can reach significant values. In such cases, the sampling frequency ($f_{dg}$) does not satisfy the requirement $f_{dg} \geq 2F_{max}$ and aliasing effect can appear.

In our results, *fd* was found to be close to Nyquist frequency ($F_{Nq}$) in many cases. This fact tells us that the effect of aliasing is present. Unless for the presence of this effect, the spectrum would have reached zero at the Nyquist frequency (in theory) or it would be close to zero due to thermal noises in the reality (Figure 7a).

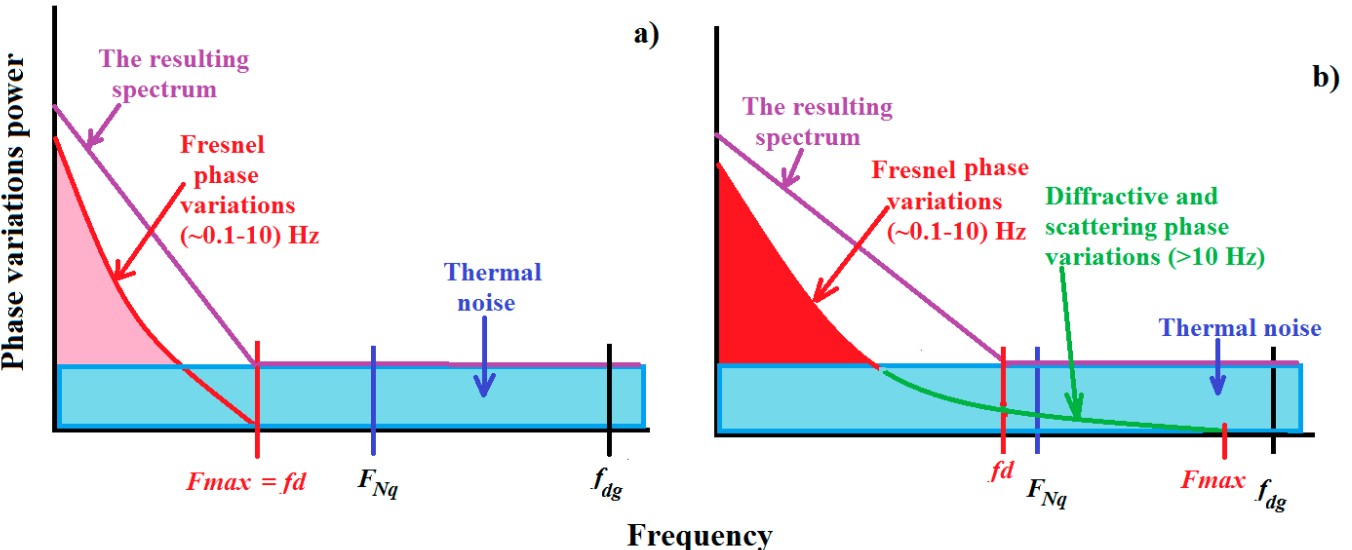

**Figure 7.** Impact of the effect of aliasing to the deviation frequency variation: in presence of refractive Fresnel variations only—panel (**a**); in presence of both refractive and diffractive phase variations—panel (**b**).

The goal of our research is to find a sensitive tool to detect as many weak small-scale ionospheric turbulences as possible from the phase variation spectra. From this standpoint, the aliasing effect brings additional opportunities to detect such turbulences by analyzing *fd* variations. This idea is illustrated in Figure 7. Two cases are considered: panel a) shows the presence of the Fresnel refractive phase variations only with frequencies no higher than 10 Hz (idealized case); panel b) displays a presence of both Fresnel refractive (<10 Hz) and high frequency diffractive and scattering phase variations (>10 Hz).

If only the Fresnel turbulences (from ~200–300 m to ~1–2 km) are considered, one may compute the correspondent frequencies of the refractive phase variations within the spectrum (Fresnel frequencies) taking into account their relative speed of drift (from ~300 m/s to ~3 km/s). The Fresnel frequencies are equal to ~ 0.1–10 Hz (i.e., *Fmax*= 10 Hz). If an exploration is focused on these ionospheric Fresnel turbulences only, the sampling frequency of 50 Hz (let alone 100 Hz) is mostly enough to guarantee that the effect of aliasing is excluded and the information about the turbulences is reliably obtained. In this case we may consider *fd = Fmax* as it is demonstrated in Figure 7a.

In a reality, however, there are always mechanisms producing the phase variations in the ionosphere at frequencies higher than 10 Hz. There is ionospheric diffraction from the small-scale ionospheric turbulences with a size of less than a hundred meters and the wave scattering from Fresnel ionospheric turbulences. Both of these ionospheric effects

produce ionospheric noise spectra coming into the other noise components (thermal noise etc.). This case is illustrated in Figure 7b. The main problem is that very little can be known about the high-frequency truncation of such phase variations' spectra, because the above-mentioned noise ionospheric variations may occur well above the sampling frequency. Hence, the noise tail of the phase variation spectra produces aliasing effects, because 100 Hz sampling frequency is probably less than double maximal frequency of the ionospheric noise variations in the spectrum. Our results where *fd* is found close to Nyquist frequency prove that 100 Hz sampling frequency is not always enough to define such high frequency ionosphere produced phase noises.

Taking all into account, we believe that small-scale ionospheric turbulences produce the effect of aliasing very often. In turn, the effect of aliasing inevitably impacts the deviation frequency behavior (look and compare Figure 7a,b). One of the goals of our research is to explore the deviation frequency variations depending on geomagnetic conditions and prove the idea that the deviation frequency is the promising tool to detect small scale ionospheric turbulences (including ones smaller than Fresnel size). Our results prove the idea successfully because we fix the reaction of *fd* variations caused by magnetic storms in a form of *fd* increasing.

## 6. Conclusions

This work presents the case study of the *fd* variations for three particular periods relating to weak magnetic storms and quiet conditions. The peculiarity of this work is that the experiment was performed at the mid-latitude station in the Northern Hemisphere. In general, this means much lower intensity of small-scale ionospheric turbulences compared to the high and low latitudes. Such an environment allowed us to better understand the *fd* reaction caused by the weak, small-scale disturbances during magnetic storms in the mid-latitude ionosphere.

The following new results were obtained:

(a) Deviation frequency apparently reacts to ionospheric disturbances triggered by magnetic storm: *fd* values are higher during geomagnetic disturbances than during the quiet periods.

(b) The distribution of observed *fd* values principally depends on the data sampling rate. The higher time rate, the wider the band of *fd* values that can be obtained, which means much better opportunities to detect weak small scale diffractive and ingattered ionospheric turbulences.

(c) *fd* variations depend on the GNSS constellation and signal features. GPS and Galileo signals show slightly higher variability in *fd* values compared to GLONASS and SBAS signals.

(d) For the 100 Hz data, the most probable values of the deviation frequency were grouped within ~30–40 Hz under quiet conditions and within ~37–48 Hz during magnetic storm. This allows us to consider the increasing of the integral number of the deviation frequencies of ~35–50 Hz as an indicator to detect weak, small-scale ionospheric disturbances.

**Author Contributions:** V.D. developed the conceptualization of this work; V.D. and Y.Y. designed the experiments; E.D. and M.S. provided the data processing, and performed the simulations and the experiments; all the authors participated in the analysis of the results; V.D. and M.S. prepared the manuscript with contributions from all the authors. All authors have read and agreed to the published version of the manuscript.

**Funding:** The work was supported by the Ministry of Science and Higher Education of the Russian Federation.

**Data Availability Statement:** All the phase variation spectra considered in the paper can be found at: Danilchuk E., Demyanov V. Fluctuation spectra. Archive of fluctuation spectra includes following sets of the data: 50 Hz of Septentrio (April 2022, September 2022), 100 Hz of Septentrio (September 2022) and 50 Hz of Javad (April 2021). https://zenodo.org/record/7539521, accessed on 16 January 2023 (alternative link is https://zenodo.org/record/7539521#.Y9dCmuD9Vkx, accessed on 16 January 2023). SYM-H data were obtained from the NASA/GSFC's Space Physics Data Facility's OMNIWeb service (https://omniweb.gsfc.nasa.gov, accessed on 16 January 2023). Kp-index data were obtained from GFZ German Research Centre for Geosciences (http://ftp.gfzpotsdam.de, accessed on 16 January 2023) [42].

**Acknowledgments:** We thank Artem Vesnin for his help in GNSS data acquisition. The study used equipment of the Center for Common Use «Angara» of ISTP SB RAS (http://ckp-rf.ru/ckp/3056/. Last access: 24 December 2022) operating under Ministry of Science and Higher Education of the Russian Federation.

**Conflicts of Interest:** The authors declare no conflict of interest.

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
