# Peer review of "An Increase of GNSS Data Time Rate and Analysis of the Carrier Phase Spectrum"

_remotesensing, doi:10.3390/rs15030792_

Round 1
Reviewer 1 Report (Previous Reviewer 1)
The authors made an essential improvement of the paper, and I believe it can be published now. The only minor issue is that there are unexplained abbreviations, like GNSS, ROTI, DROTI. The abbreviation must be explained at the first use (even if the authors consider them commonly-known).
Author Response
Reviewer#1
The authors made an essential improvement of the paper, and I believe it can be published now. The only minor issue is that there are unexplained abbreviations, like GNSS, ROTI, DROTI. The abbreviation must be explained at the first use (even if the authors consider them commonly-known).
Reply
Thank you very much for your positive opinion and for you time and useful critics. According to your remark, we added the abbreviations explanations in the text at their first use (the modified parts of text are marked with blue color in the improved manuscript version).
Reviewer 2 Report (Previous Reviewer 2)
After reading through the revision, I think I don’t have any more criticism on the manuscript. It’s a green light to be published now from my side.
Author Response
Reviewer#2
After reading through the revision, I think I don’t have any more criticism on the manuscript. It’s a green light to be published now from my side.
Reply
Thank you very much for your positive opinion and for you time and useful critics.
This manuscript is a resubmission of an earlier submission. The following is a list of the peer review reports and author responses from that submission.
Round 1
Reviewer 1 Report
The authors analyze the fluctuation spectra of GNSS carrier phase measured by ground-based receivers. To this end, the authors evaluate the so-called deviation frequency (fd), i.e. the frequency, where the fluctuation spectrum reaches the noise floor. The authors pose a question about the influence of the sampling rate on the informative part of the spectrum.
Although the design of the study definitely makes sense, I have some doubts regarding the presented results. The main problem is the definition of fd. The authors state:
"The deviation frequency (fd) value is found as a point in the phase variations spectra at which the spectral slope shallows to the near zero inclination. Figure 1 illustrates fd that divides the informative part of the phase variations spectra and phase uninformative noise."
But in Figure 1 we don't see any clear point of the spectral slope change, because the supposed fd is too close to the Nyquist frequency. If we take a look at the previous paper of the authors (https://www.mdpi.com/2072-4292/13/24/5017)
we find just two examples of the definition of fd in Figure 2. The figure is accompanied by a formal description of the procedure of finding fd. However, I notice the following problems. The fd in both examples is again located very closely to the Nuquist frequency. But a well-known fact is that the aliasing effect results in the overestimate of the spectrum at this frequency at least twice. So, at a qualitative level, what we see here looks very much like the effect of aliasing rather than the spectrum reaching the noise floor.
Another question is about the spectrum shape. It is hard to believe that all the fluctuation spectra obey the same power law. A general rule for such spectra is that in different ranges they have different slopes.
There is one more point that is not clear to me. System like GPS, GLONASS, QZSS modulate the signal with navigation message at 50 Hz rate. In other observation types, like radio occultation, externally supplied navigation bit sequences are used for the demodulation. How is the demodulation performed here?
Finally, I think that the paper in its present form should be rejected. But the authors must be encouraged to take their time, in order to update and re-submit the paper. In particular, much more examples of fluctuation spectra for different conditions must be presented.
There are some other minor things:
Table 1. This is a table. Tables should be placed in the main text near to the first time they are cited.
Fill in the table caption, instead of the one inherited from the template.
Eq. (1): sin sin should be probably just sin, and cos cos should be just cos.
However, I think that first the major questions are to be clarified.
Reviewer 2 Report
This technical note studied GNSS-based ionospheric disturbance detection using the deviation frequency as the sensitivity threshold from carrier phase spectrum. A positive correlation was found between deviation frequency and geomagnetic activity level from multi-constellation multi-signal GNSS data analysis in Irkutsk, Russia (52° N, 104° E). Based on the experiment results, the proposed method is applicable to detect small-scale ionospheric irregularities with a deviation frequency of ~35–50 Hz. Overall, the manuscript is well written in background introduction, methodology, experiment setup and result analysis. Before proceeding to publication, I encourage the authors to address my following comments:
1. [Line 50-52] “As TEC-based data is a product of inter-frequency ionosphere-free combination, it can be degraded. First, there is significant difference between the L2P(Y) and L2C-derived TEC.”
o The phrase “it can be degraded” can be confusing. Are you only discussing GPS-derived TEC? If so, does the issue still exist when using L1 and L5 frequency band GPS data to derive TEC?
2. [Line 75-77] “Stochastic techniques report the normally distributed carrier phase noise of 0.002 m, while the code range noise is of (0.5-0.8) m [19].”
o The stochastic measurement uncertainty of carrier phase and code range depends on receiver software and hardware systems. For smartphone grade GNSS receivers, the uncertainty level can be worse than the number you provide. Please specify which type of receivers you are referring to with the number (e.g., geodetic grade, space weather monitoring type)
3. [Table 2] I see a large empty space in the table covering most of the Javad Delta-G3T section and a small bland region in the Septentrio PolaTx5 April 13 and April 16 region. Are they expected?
4. [Conclusion] Only one site is chosen in this experiment. Is it general enough to summarize the result in the Northern hemisphere mid-latitude region? If another one or two experiment in other mid-latitude region can be included, the conclusions will be more solid.